# LDH-A Promotes Metabolic Rewiring in Leucocytes from the Intestine of Rats Treated with TNBS

**DOI:** 10.3390/metabo13070843

**Published:** 2023-07-12

**Authors:** Belen Mendoza-Arroyo, Martha Cecilia Rosales-Hernández, Judith Pacheco-Yépez, Astrid Mayleth Rivera-Antonio, Yazmín Karina Márquez-Flores, Luz María Cárdenas-Jaramillo, Aldo Arturo Reséndiz-Albor, Ivonne Maciel Arciniega-Martínez, Teresita Rocío Cruz-Hernández, Edgar Abarca-Rojano

**Affiliations:** 1Sección de Estudios de Posgrado e Investigación, Escuela Superior de Medicina, Instituto Politécnico Nacional, Plan de San Luis y Díaz Mirón s/n., Ciudad de México 11340, Mexico; bmendozaa1200@alumno.ipn.mx (B.M.-A.); jpachecoy@ipn.mx (J.P.-Y.); trcruz@ipn.mx (T.R.C.-H.); 2Laboratorio de Biofísica y Biocatálisis, Sección de Estudios de Posgrado e Investigación, Escuela Superior de Medicina, Instituto Politécnico Nacional, Plan de San Luis y Díaz Mirón s/n., Ciudad de México 11340, Mexico; mrosalesh@ipn.mx (M.C.R.-H.); astridmriveraa@gmail.com (A.M.R.-A.); 3Departamento de Farmacia, Escuela Nacional de Ciencias Biológicas, Campus Zacatenco, Instituto Politécnico Nacional, Av. Wilfrido Massieu s/n Col. Zacatenco, Ciudad de México 07738, Mexico; ymarquez@ipn.mx; 4Coordinación de Ciencias Morfológicas, Escuela Superior de Medicina, Instituto Politécnico Nacional, Plan de San Luis y Díaz Mirón s/n., Ciudad de México 11340, Mexico; luzma9697@gmail.com; 5Laboratorio de Inmunidad de Mucosas, Escuela Superior de Medicina, Instituto Politécnico Nacional, Plan de San Luis y Díaz Mirón s/n., Ciudad de México 11340, Mexico; aresendiza@ipn.mx (A.A.R.-A.); iarciniega@ipn.mx (I.M.A.-M.)

**Keywords:** epithelium, leukocytes, lamina propria, TNBS-rats, lactate, LDH-A

## Abstract

Although the aetiology of inflammatory bowel diseases (IBDs) is still unknown, one of their main characteristics is that the immune system chronically affects the permeability of the intestinal lamina propria, in turn altering the composition of the microbiota. In this study, the TNBS rat model of colitis was used because it contains a complex inflammatory milieu of polymorphonuclear cells (PMN) and lymphocytes infiltrating the lamina propria. The aim of the present study was to investigate six dehydrogenases and their respective adaptations in the tissue microenvironment by quantifying enzymatic activities measured under substrate saturation conditions in epithelial cells and leukocytes from the lamina propria of rats exposed to TNBS. Our results show that in the TNBS group, an increased DAI score was observed due to the presence of haemorrhagic and necrotic areas in the colon. In addition, the activities of G6PDH and GADH enzymes were significantly decreased in the epithelium in contrast to the increased activity of these enzymes and increased lactate mediated by the LDH-A enzyme in leukocytes in the lamina propria of the colon. Over the past years, evidence has emerged illustrating how metabolism supports aspect of cellular function and how a metabolic reprogramming can drive cell differentiation and fate. Our findings show a metabolic reprogramming in colonic lamina propria leukocytes that could be supported by increased superoxide anion.

## 1. Introduction

Inflammatory bowel diseases (IBDs), namely ulcerative colitis (UC) and Crohn’s disease (CD), are chronic immune-mediated conditions [1]. Despite some shared characteristics, these forms can be distinguished by differences in genetic predisposition, risk factors, and clinical and histological features [2]. It is widely accepted that UC is triggered by events that perturb the mucosal barrier, e.g., disruption of epithelial tight junctions increases intestinal permeability [3], alters the healthy balance of the gut microbiota, and abnormally stimulates gut immune responses [1]. The 2,4,6-trinitrobenzenesulfonic acid (TNBS) colitis model is one of the models used in experimental studies of inflammatory bowel disease (IBD) since inflammation induced with TNBS mimics several features of CD [4,5]. It has been documented that 2,4,6-trinitrobenzenesulfonic acid (TNBS) reacts with certain groups of amino acids in the intestinal mucosa and colonic bacterial proteins, making them immunogenic through a process called haptenation [6]. TNBS haptenizes autologous colonic proteins with a trinitrophenyl moiety and induces interleukin 12 (IL-12)-mediated proinflammatory transmural colitis and TH1 lymphocyte infiltration [6]. Therefore, we selected TNBS over other models of colonic inflammation because of its early potent recruitment of PMN cells [5,7] as well as CD4^+^ T cells [8]. This model is useful for studying the metabolic role of colonocytes and immune cells in the mucosa. A previous study found that oral antibiotic exposure during the first five years of life was associated with an increased risk of inflammatory bowel disease (IBD), and repeated antibiotic exposure increased the risk estimates [9]. It was proposed that IBD, Crohn’s disease, and ulcerative colitis, are associated with an increased risk of symptomatic *Clostridium* difficile infection [10]. Histologic observations of subepithelial inflammation implicate impaired epithelial barriers as a damaging factor in UC. This is through either altered or impaired secretion (e.g., of antimicrobial peptides, damage-associated molecular patterns, or mucus) or physical defects (e.g., from disruption of epithelial tight junctions that increase intestinal permeability) [8] allowing increased entry of luminal antigens or defective regeneration or detoxification [11]. On the other hand, interesting results have suggested that the epithelium of the colon contributes to immune functions that maintain homeostasis by shaping the microbiota to be beneficial [12]. A recent study described that colitis from TNBS is characterised by tissue damage and uncontrolled inflammation. Neutrophils and other inflammatory cells play a primary role in disease progression by acutely responding to direct and indirect insults to tissue injury and promoting inflammation through the secretion of inflammatory cytokines and proteases. Vascular endothelial growth factor (VEGF) is a ubiquitous signalling molecule that plays a key role in maintaining and promoting cell and tissue health and is dysregulated in UC. Recent evidence suggests a role of VEGF in mediating inflammation [13]. A reduced presence of neutrophils was found to be associated with decreased pro-inflammatory cytokines (including TNFα, IL-1β, and IL-6) and myeloperoxidase (MPO) [12]. The colonic epithelium is continually renewed by colonic stem cells located at the base of intestine glands, termed the crypts of Liberkühn. Importantly, the cells located at the base of crypts obtain energy through aerobic glycolysis, which is characterised by the conversion of glucose into lactate in the presence of oxygen once the colonic epithelium has matured [14,15]. Evidence shows that the butyrate (a metabolite of microbiota) actives PPAR-γ (peroxisome proliferator-activated receptor-γ) in epithelial cells differentiated under basal conditions [16]. PPAR-γ is a nuclear receptor that activates fatty acid metabolism to drive the metabolism of surface colonocytes toward mitochondrial β-oxidation of fatty acid, which is important for maintaining epithelial hypoxia [17]. This results in mitochondrial oxidation of median and short-chain fatty acids as well as oxygen consumption through oxidative phosphorylation [17,18,19]. The dysbiosis generates a shift in reprogramming metabolic epithelium with the loss of epithelial hypoxia, thereby driving expansions of aerobic bacteria, emanating from the epithelial surface, and triggering the inflammatory response, anaerobic glycolysis, lactate release, and other nitrogen and oxygen radicals that promote an environment proinflammatory [14]. Moreover, unlike immune system cells, which, when activated, require a metabolism characterised by low oxygen consumption, high glucose consumption and high lactate release (Warburg effect) under conditions of aerobic glycolysis, from glycolysis, part of the glucose is directed via the pentose pathway to the immune system to synthesise nucleotides for the formation of pro-inflammatory interleukins [20,21,22].

The aim of our study was to measure the enzyme activities of six dehydrogenases from substrate to saturation conditions. The measured activities of the respective enzymes and their adaptations in response to inflammation stimuli serve as direct indicators for the configuration comprising the pathways of primary carbohydrate metabolism, including glycolysis, the pentose phosphate pathway (PPP), and the tricarboxylic acid (TCA) cycle. In addition, we studied superoxide anion because it has been proposed that in some leukocytes, it initiates signal transduction by increasing the production of superoxide anion (O^2−^) and hydrogen peroxide (H_2_O_2_) as second messengers [23]. Our results show that in the TNBS group, the following were detected in the colon: weight loss, less water and food consumption, DAI increase, haemorrhagic and necrotic areas, morphological changes, and inflammation. We also found increased glucose-6 phosphate dehydrogenases (G6PH), glyceraldehyde 3-phosphate dehydrogenases (GADH), lactate dehydrogenase (LDH), and succinate dehydrogenases (SDH) in leukocytes. These measurements represent the amount of active enzyme in the analysed samples as well as an increase in the metabolite lactate produced by LDH, which is an isoenzyme and consists of LDH, LDH-A, LDH-B, and LDH-C subunits encoded by four different genes, indicating that the glycolysis could define the specialised architecture of a microenvironment.

## 2. Materials and Methods

### 2.1. Experimental Animals

A total of 12 female Wistar rats (weight, 280–300 g; age, 10–12 weeks) were used in this study. All animals were kept on a 12 h light/dark cycle (lights on at 6:00 a.m.) at room temperature (RT). They were fed a rodent diet (Lab Diet 5013) and water ad libitum. The current protocol was developed based on the ARRIVE guidelines for reporting animal research [23] and was approved by the Research Committee of the Escuela Nacional de Ciencias Biológicas (CEI-ENCB) (ENCB/CEI/002/2022). The animals were handled following the Mexican Federal Regulations for animal experimentation and care (Regulation-062-ZOO-1999), Ministry of Agriculture, México City, Mexico.

### 2.2. Induction of Colitis in Rats with TNBS

Colitis was induced according to a previously described procedure [6] modified by Calva Candelaria [24], in which colonic injury was induced. The animals were randomly divided into two groups (n = 6): (i) ethanol group (EtOH) with 50% ethanol enema and (ii) TNBS group (TNBS), which was administered TNBS one day (Sigma Aldrich, Merck, St. Louis, MO, USA). The rats in the TNBS group were lightly anaesthetised with ether and administered TNBS 0.25 mL of 10 mg in solution saline 0.9% with 50% ethanol enema. Before the induction of experimental colitis, there was a 12 h period without food. Once anaesthetised, the animals were administered with a medical-grade polyurethane catheter (2 mm) so that the tip entered smoothly and without resistance 8 cm proximal to the anal verge. After this, the rodents are kept for 30–60 s in the Trendelenburg position (head down) to prevent fluid expulsion and ensure even distribution of TNBS. The rats were checked daily to identify changes in behaviour, water and food consumption, and parameters of the DAI scale: body weight, consistency, and presence of blood in the stool.

### 2.3. Assessment of Colitis

The rats were sacrificed 48 h after the induction of ulcerative colitis. From each animal, 10 cm of the colon was removed. Samples were slightly cleaned with physiological saline solution to remove faecal residues and then weighed [25]. The disease activity index (DAI) was obtained considering the assigned scores based on some clinical features of the colon and loss of body weight (score 0–3). The presence of adhesions (score 0–2), stool consistency (score 0–3), and the presence of blood in the stool (score 0–3) were evaluated according to the criteria of Márquez-Flores et al. 2000 [26]. Samples of the colon were collected and frozen (−70 °C) for the measurement of biochemical parameters [24].

### 2.4. Tissue Collection

Upon sacrifice, we administered nasal isoflurane at a lethal dose (PISA, Mexico City, Mexico) (100 mg/kg). Blood was obtained with a cardiac puncture (5 mL–9 mL), and serum was obtained with centrifugation at 1660× *g* for 7 min at 4 °C. The serum samples were stored at −20 °C until use. After exsanguination, the large intestines were dissected, and colon segments were cut (1 cm), fixed with 4% paraformaldehyde for 24 h, and processed for paraffin embedding. The rest of the intestines were separated and used for enzymatic activity, lactate determination, and superoxide anion quantification.

### 2.5. Haematoxylin and Eosin Staining

Colon sections (5 µm thick) were generated with a microtome (Rotatory Microtome; Leica Microsystems GmbH, Wetzlar, Germany), placed on coverslips, and stained with H&E. After deparaffinization with xylol and rehydration in a descending alcohol gradient, the samples were immersed in Harris haematoxylin solution (Sigma-Aldrich; Merck, St, Louis, MO, USA) and incubated for 2 min at room temperature. After incubation in an eosin solution for 2 min at RT (Sigma-Aldrich, Merck, St. Louis, MO, USA), the samples were then washed with distilled water. Finally, the sections were dehydrated and mounted with Entellan^®^ (Sigma-Aldrich, Merck, St. Louis, MO, USA).

### 2.6. Epithelial Cell Isolation

The intestine fragments (5 ± 1 cm, approx.) were incubated in RPMI-1640 medium (Sigma-Aldrich, Merck, St. Louis, MO, USA) with 1 mM dithiothreitol (Thermo Fisher Scientific, San Diego, CA, USA) and 1.5 mM EDTA (Sigma-Aldrich, Merck, St. Louis, MO, USA) with continuous stirring at 415× *g* for 30 min at 37 °C. The cell suspension was passed through organza to remove mucus and centrifuged at 415× *g* for 10 min at 4 °C. The pellet was suspended in 15 mL of RPMI-1640 medium (Thermo Fisher Scientific, San Diego, CA, USA), passed through an organza filter, and washed twice with 15 mL of RPMI-1640 medium followed by centrifugation at 415× *g* for 10 min at 4 °C. The washed pellet was suspended in 40% Percoll^®^ (Sigma-Aldrich, Merck, St. Louis, MO, USA) and centrifuged in a Percoll discontinuous gradient at 75% at 1160× *g* for 30 min at 25 °C. Epithelial cells were recovered from the top, and the lamina propria leukocytes were recovered from the interface between 40% and 75%. The cells were washed with PBS and centrifuged, as previously mentioned. The cells were then resuspended in RPMI-1640 medium. The purity of the samples was analysed with light microscopy based on the normal morphology of epithelial cells. The cells were resuspended in RPMI-1640 medium. The vials were stored at −80 °C.

### 2.7. Protein Quantification

The cells obtained were sonicated with 1 mL of 50 mM trizma base cold (Sigma-Aldrich, Merck, St. Louis, MO, USA) at pH 7.8 and 1 mM ethylenediaminetetraacetic acid (EDTA) (Sigma-Aldrich, Merck, St. Louis, MO, USA) to lyse the cells and subsequently centrifuged at 10,000 rpm for 10 min at 4 °C. In a 96-well plate, 10 µL of cell supernatant from each test group was placed in triplicate, and 90 µL of Milli-Q water was added. The methodology suggested in the Protein Determination Kit (Cayman Chemical, Ann Arbor, MI, USA) was used.

### 2.8. Assessment of Enzymatic Activities in Cellular Fractions

To determine enzyme activity, we measured the amount of formed product over time and the initial velocity (V_0_ catalytic rate) at optimal and constant assay conditions, including saturating substrate and cofactor levels (V0 = Vmax) [27] (0–14 for epithelium) to (0–45 min by leukocytes). This measurement of enzymatic activities in this model may serve as direct indicators for the configuration that comprises the pathway of primary carbohydrate metabolism, including glycolysis, the pentose phosphate pathway (PPP), and the tricarboxylic acid (TCA) cycle [27]. Briefly, for the enzyme activity assays, enzyme-specific buffers were prepared as follows. For glucose 6 phosphate dehydrogenase (G6PD), isocitrate dehydrogenase (IDH), lactate dehydrogenase (LDH), and glutamate dehydrogenase (GDH) assays, the buffer was 7.5 mM G6P (D-glucose 6-phosphate solution; Sigma-Aldrich, Merck, St. Louis, MO, USA), 0.8 mM NADP^+^ (β-nicotinamide adenine dinucleotide phosphate hydrate; Sigma-Aldrich, Merck, St. Louis, MO, USA), and 4 mM Mg^+^CL (magnesium chloride hexahydrate, JT Baker Fisher Scientific, Phillipsburg, NJ, USA) for the analysis of glucose 6P dehydrogenase (G6PD) activity. The buffer was 7.5 mM G6P, 0.8 mM NAD^+^ (β-nicotinamide adenine dinucleotide hydrate; Sigma-Aldrich, Merck, St. Louis, MO, USA), and 4 mM (Mg^+^CL^−^), 2.5 mM, glyceraldehyde-3-phosphate (GAPD) (DL-glyceraldehyde 3-phosphate diethyl acetal barium salt; Santa Cruz Biotechnology, Dallas, TX, USA) and 0.8 mM NAD^+^ (β-nicotinamide adenine dinucleotide hydrate (Sigma-Aldrich, Merck, St. Louis, MO, USA) for the analysis of GAPD activity. The buffer was 150 mM pyruvate (sodium L-lactate; Sigma-Aldrich, Merck, St. Louis, MO, USA) and 0.8 mM NADH^+^ (β-nicotinamide adenine dinucleotide phosphate hydrate (Sigma-Aldrich, Merck, St. Louis, MO, USA) for the analysis of lactate dehydrogenase (LDH) activity (sodium L-lactate; Sigma-Aldrich Merck, St. Louis, MO, USA). The buffer was isocitric acid (4 M) trisodium salt hydrate (Sigma-Aldrich, Merck, St. Louis, MO, USA) for the analysis of isocitric dehydrogenase (IDH). The buffer was 0.8 mM NAD^+^ (β-nicotinamide adenine dinucleotide hydrate (Sigma-Aldrich, St. Louis, MO, USA) for the analysis of IDH activity. The buffer was 0.4 M and 0.8 mM NAD+ for the analysis of glutamate dehydrogenase (GDH) (L-glutamic acid monosodium salt hydrate; Sigma-Aldrich, Merck, St. Louis, MO, USA). For the enzyme activity assays of succinate dehydrogenase (SHD) (sodium succinate dibasic; Biovision, Milpitas, CA, USA), the buffer was 50 mM Tris Base buffer pH 7.5, 0.1 M Tris-HCl (Tris hydrochloride; US Biological, Boston, MA, USA) buffer pH 8.0, 60 mM sodium succinate 0.8 mM FAD+ (flavin adenine dinucleotide disodium salt hydrate; Sigma-Aldrich, Merck, St. Louis, MO, USA), and 2 mM DCPIP (Sigma-Aldrich, St. Louis, MO, USA). Negative control reactions were performed in the absence of substrate. This was performed in triplicate with all cell groups for subsequent readings at 340 nm and 600 nm (SDH only) using the Multiskan Sky Thermo Scientific microplate spectrophotometer (San Diego, CA, USA) every 2 min for 46 min. After the mean enzyme activity per μg of protein was calculated using a calibration curve with increasing concentrations of NADPH, NADH, and FADH (0.625, 1.5, 2.5, 5, 7.5, 10 and 12.5 mmol) dissolved in 150 µL of Tris base (Sigma-Aldrich, Merck, St. Louis, MO, USA) in each well in triplicate, the samples were read using the spectrophotometer at 340 nm.

### 2.9. Plasma Lactate Measurement

Two millilitres of blood were placed in tubes with EDTA (ethylenediaminetetraacetic acid tetrasodium salt dihydrate; Sigma-Aldrich, Merck, St. Louis, MO, USA), and the other 2 mL was placed in a cell preparation tube (Vacutainer^®^ CPT™ Mononuclear Cell Preparation Tube, BD Biosciences, Franklin Lakes, NJ, USA). The intestine was macerated, and epithelial cells and lamina propria lymphocytes were obtained. All the cells previously obtained were sonicated for 5 s. Subsequently, lactate was determined in the sonicated samples using the kit L-Lactate Standard (LAC CAL, Randox. Laboratories, Crumlin, UK).

### 2.10. Quantification of Superoxide Anion Using Electron Paramagnetic Resonance

Electronic paramagnetic resonance (EPR) was used to quantify the superoxide anion. The experiment was carried out under the following conditions: at 50 mg of each sample from the intestine, 200 μL of radical scavenger CMH (1-hydroxy-3-methoxycarbonyl-2,2,5,5-tetramethylpyrrolidine) (NOXYGEN, Elzach, Germany) at 50 mM dissolved in Krebs buffer (99 mM NaCl, 4.69 mM KCl, 2.5 mM CaCl_2_ × 2H_2_ O, 1.2 mM MgSO_4_ × 7H_2_ O, 2.5 mM NaHCO_3_, 1.03 mM KH_2_ PO_4_, 5.6 mM D-glucose, 20 mM Na-HEPES) with sodium diethyldithiocarbamate trihydrate (5 μM DETC) (NOXYGEN, Elzach, Germany), and deferoxamine methanesulfonate salt (25 μM DF) (NOXYGEN, Elzach, Germany) was added. The CMH preparation and the samples were processed inside a sealed chamber with a nitrogen atmosphere. The samples containing CMH were incubated for 30 min at 37 °C with constant shaking. Next, the samples were centrifuged at 12,000× *g* for 10 min, and the supernatant was extracted in 50 μL capillaries (Corning Inc., Corning, NY, USA) inside the nitrogen chamber. Each of the capillaries was sealed on both sides to then process the EPR measurements. The measurements of O^2•−^ formation were performed using a Bruker e-scan spectrometer (Biospin, Billerica, MA, USA) with the following recorded parameters: magnetic field 3490 G, window: 60 G, amplitude: 1.5 G, gain: 56.4, power: 0.0155 mW, frequency: 9.749 GHz. WinEPR software was used to analyse the results (area values). The data are represented as the mean ± SD, and the plotting and statistical analysis (*p* < 0.0) using 2-way ANOVA were performed with GraphPad Prism 6 software (6.04 version, Palo Alto, CA, USA). 

### 2.11. Immunohistochemistry LDH-A in TNBS-Induced Colitis in Rats

Colon samples (5 μm cuts) were deparaffinised and hydrated with PBS. We performed antigenic recovery in citrate buffer (10 mM, 0.05% Tween-20) at 90 °C for 10 min. Then, the colon samples were chilled and washed with citrate buffer for 20 min. Endogenous peroxidase was blocked by inhibition with 3% H_2_O_2_ in PBS for 30 min. A block with FBS at 3% was made. The samples were incubated with the primary antibody anti-LDH-A peroxidase 1:100 (Genetex, Irvine, CA, USA) overnight at 4 °C in a humid chamber. The diaminobenzidine kit was added at a 1:9 dilution (Vector Laboratory, Newark, CA, USA), subsequently counterstained with haematoxylin (1:9), washed with tap water, and then washed with distilled water. Finally, the samples were dehydrated in alcohol and xylene and analysed with microscopy (Nikon, Eclipse E600, Melville, NY, USA).

### 2.12. Statistical Analysis

The experimental assays were repeated for ≥3 independent assays (n = 6 rats per group). The data were expressed as the mean ± SD, and multiple comparisons between groups were analysed using one *t*-student impair. Statistical analyses were performed with the diaminobenzidine med kit using GraphPad Prism Version 9 software (GraphPad Software, Inc., Palo Alto, CA, USA). *p* < 0.001 was considered to indicate a statistically significant difference.

## 3. Results

### 3.1. A Rise in the Disease Activity Index (DAI) in TNBS-Induced Colitis in Rats 

Acute colonic inflammation was induced one day using intrarectal administration with TNBS. Then, 48 h after the administration, the rats were euthanised and showed significant weight loss and less water and food consumption, increase in DAI index, adhesions, haemorrhage, and necrotic areas (Figure 1 and Appendix A). On the day of sacrifice, the EtOH control group weighed 295 g (Figure 1A) and consumed 68 mL of water (Figure 1B) and 36 g of food (Figure 1C); the group treated with TNBS had a final weight of 257 g, reflecting a reduction close to 9% (*p* < 0.0401) (Figure 1A), and consumed 32 mL of water (*p* < 0.001) (Figure 1B) and 19 g of food (*p* <0.001) (Figure 1C). DAI was significantly increased in the TNBS group (*p* < 0.016) (Appendix A and Figure 1D). Furthermore, TNBS instillation induced inflammatory zone post-mortem adherences (adherence are scar-like bands of tissue) (*p* < 0.007) in the descending colon region (score Figure 1D) as well as ulcers and haemorrhage and necrotic areas (score Figure 1E,F).

### 3.2. Morphological Changes in TNBS-Induced Colitis in Rats Rats

Haematoxylin and eosin staining in rats treated with ethanol showed slight variations in the epithelium, and there was a scare infiltrate under the epithelium (asterisk) and at the bottom of the intestinal crypts (Figure 2A). In the colon epithelium of TNBS-rats, areas with flattened epithelium are observed, and there was oedema in the lamina propria and submucosal layer (arrow). The intestinal glands were separated because of the oedema (asterisk). A clear presence of goblet cells was evident (asterisk) (Figure 2B).

### 3.3. TNBS-Induced Colitis in Rats Modifies the Enzymatic Activity in Epithelial Cells 

The results for the enzymatic activities of the epithelial cells were obtained by analysing the concentration of NADPH, NADH, and FADH produced/µg of protein. During the evaluation of initial enzymatic activity, we did not find a significant increase in the TNBS groups with respect to EtOH except for succinate dehydrogenase (SDH) (0.16 ± (0.0007) (0.11 ± (0.0001) (*p* ≤ 0.0001) (Appendix A). We compared the values of the enzymatic activities with respect to the time as evaluated in the EtOH and TNBS groups. We observed that all the enzymes presented changes in enzymatic activity that were statistically significant. However, the activity of the enzyme glucose-6-phosphate dehydrogenase (G6PDH) in the TNBS group decreased 10 min after adding the substrate during the enzymatic activity assay (*p* ≤ 0.0002) (Figure 3A). Unlike in the TNBS group, isocitrate dehydrogenase (IDH) activity increased during the first 24 min (*p* ≤ 0.0001) (Figure 3B). For the glutamate dehydrogenase (GDH) enzyme, activity in the TNBS group increased in the first 10 min of reading in the spectrophotometer; however, this activity was observed in the EtOH group from minute 12 (*p* ≤ 0.0001) (Figure 3C). In our model of inflammation with TNBS, we found no increase in the activity of glycolysis enzymes, only an increase in succinate dehydrogenase. Interestingly, measurement of this dehydrogenase could follow the electronic activity of flavoproteins, e.g., respiratory complex II. In relation to these results, we think it is likely that an anapletoric source other than glucose promotes the activity of these enzymes (Appendix A and Figure 3).

### 3.4. TNBS-Induced Colitis in Rats Modifies the Enzymatic Activity of Leukocytes in the Lamina Propria

We evaluated the activity of the dehydrogenases in the leukocytes of the lamina propria, namely G6PDH, GAPDH, lactate dehydrogenase (LDH), IDH, SDH, IDH, and GDH, obtained from rats treated with a 50% ethanol enema (EtOH group) or with TNBS. The results obtained for the initial activity of G6PDH showed that the TNBS group increased (0.035 ± 0.007) µg/NADPH/µg protein compared to the EtOH control group (0.019 ± 0.001) (*p* ≤ 0.01). For GAPDH, the activity in the TNBS group increased (0.657 ± 0.246) compared with EtOH group (0.232 ± 0.016) (*p* ≤ 0.04). For LDH in the TNBS group, the concentration in the initial activity increased (0.448 ± 0.119) compared to the EtOH group (0.185 ± 0.005) (*p* ≤ 0.01). IDH decreased in the TNBS group (0.018 ± 0.009) compared with the EtOH group (0.023 ± 0.006) (*p* ≤ 0.447). GDH decreased (0.017 ± 0.015) in the TNBS group vs. the EtOH group (0.042 ± 0.010) (*p* ≤ 0.082). The SDH-TNBS group increased (0.044 ± 0.111) compared to (0.011 ± 0.0008) the EtOH group (*p* ≤ 0.006). Unlike the other dehydrogenases, SDH contains FAD^+^. These statistical differences are shown in Table 1. The activity of the enzymes was analysed at 45 min, and it was observed that all the enzymes presented statistically significant differences; however, the activity of the most notable enzyme that increased was LDH, and the concentration of µg NADH/µg protein increased in the first two minutes, compared to the EtOH group (*p* ≤ 0.0001) (Figure 4A). For IDH, the concentration of µg NADH/µg protein in the TNBS group decreased during the first 4 min and remained constant over time, unlike what was observed for the EtOH group (*p* ≤ 0.0001) (Figure 4B). For GDH, the TBNS group decreased in the concentration of µg of NADH/µg of protein with respect to the EtOH group, which was statistically significant (*p* ≤ 0.0001) (Figure 4C). Overall, the increased enzyme activities of G6PDH, G3PDH, and LDH, which are the enzymes responsible for the pentose phosphate (G6PDH) and glycolysis (G3PDH) and (LDH) pathways, as well as the enzymatic measurement over time of LDH, indicate that these metabolic pathways are active in the leukocytes of TNBS-treated rats.

### 3.5. LDH-A Presence in the Colon Inflammatory Infiltrate of TNBS-Induced Colitis in Rats

The immunohistochemistry of LDH-A showed positive labelling in the epithelium of ethanol- and TNBS-treated rats (arrows A and B). LDH-A labelling was also evident in cells of the inflammatory infiltrate, and LDH-A positivity was detected in neurons of the myenteric plexus (Auerbach) (box increase to 40 X) (A). In Figure 5B, a positive label is observed in the cellular inflammatory infiltrate (box increase to 40 X). Although we measured LDH activity in this study, we thought that, according to the literature, the LDH-A isoform might be present in the leukocytes of TNBS-treated rats further promoting lactate release and effect, and we confirmed the presence of LDH-A (Figure 5 and Figure 6).

### 3.6. Increase in Lactate Concentration in Lamina Propria Leukocytes of TNBS-Induced Colitis in Rats

We measure plasma lactate concentration to rule out statistical differences between EtOH- and EtOH+TNBS-treated rats. The release of lactate in lymphocytes in the blood, plasma, colon epithelial cells, lamina propria leukocytes, and the macerated colon was measured. Our results demonstrated that lactate release was similar in lymphocytes present in blood, plasma, colon epithelial cells, and intestines macerated in the EtOH and TNBS groups (Figure 6A). However, lamina propria leukocytes in the TNBS group showed a significant increase in lactate concentration (0.7313 mmol/L) unlike in the EtOH group (0.561 mmol/L) (* *p* = 0.001) (Figure 6B). Although we measured LDH activity in this study, we thought that, according to the literature, the LDH-A isoform might be present in the leukocytes of TNBS-treated rats further promoting lactate release and effect, and we confirmed the presence of LDH-A (Figure 5 and Figure 6).

### 3.7. Epithelial Cells and Leucocytes Increase the Levels of Superoxide Anion in the Colon of TNBS-Induced Colitis in Rats

The superoxide anion (O^2^•−) level was determined using EPR. Epithelial cells and leukocytes were used in the EtOH and TNBS groups. The nitrogen atom of the compound 1-hydroxy-3-methoxycarbonyl-2,2,5,5-tetramethylpyrrolidine (CMH) trapped free radicals with EPR, and it was observed that all the samples presented the same triple signal spectrum, suggesting the presence of superoxide anions (Figure 7A). Once the spectra were obtained, the area under the curve was quantified, which is directly proportional to the superoxide anion present in the samples (Figure 7B). The results showed that the epithelial cells of the EtOH group presented an area under the curve of 5790.966 AU ± 218.324 AU, while the epithelial cells of the TNBS group presented an area under the curve greater than 7260.077 AU ± 166.010 AU. The leukocytes of the EtOH group presented an area under the curve of 8364.411 AU ± 144.651 AU, while the leukocytes of the TNBS group presented an area under the curve of 8452.077 AU ± 169.015251 AU. The epithelial cells of the TNBS group significantly increased superoxide anion levels with respect to the epithelial cells of the EtOH group (Figure 7B).

## 4. Discussion

Our study analysed colonic damage parameters and colon histological changes. Also, the activities of glycolytic and Krebs enzyme activities in epithelial cells and leukocytes from the lamina propria were quantified and LDH distribution and LDH-A quantification were determined. Finally, we analysed the O_2_^−^, a product of oxidative stress in epithelial and leucocytes. The morphological analysis of the TNBS group showed loss of mucosa structure, epithelial barrier disruption, presence of leukocyte infiltrate, and oedema and damaged crypts. Previous studies demonstrated the presence of oedema, cellular infiltrate, and mucosal injury in rats treated with TNBS interruption of intestinal mucosa [28]. Additionally, the work by Martins Goncalves [29] demonstrated that rats with colitis induced with TNBS present tissue damage, ulcerations and a dense inflammatory infiltrate, deformed crypts, and oedema that caused loss of goblet cells. Our results are like to those of previous reports [29].

Additionally, our result for GADPH enzyme activity in the epithelial cells from the TNBS group agree with another study, in which GADPH activity was measured in colonic epithelial crypt cells (CECs) from the inflamed mucosa of IBD patients. The authors observed that oxidation of GAPDH active-site thiols and subsequent inhibition of enzyme activity were consistent observations in preparations of CECs from the inflamed mucosa of Crohn’s disease [30]. The decrease we found in our study for G6PDH activity in the TNBS group is consistent with this observation. However, in this study, the significant increase in G6PDH and GADPH enzyme activity in the TNBS group for the lamina propria leukocytes could agree with a study proposed for macrophages, in which the cellular requirement for NADPH exceeded nucleotide biosynthesis; this is why Ribulose (Ru5P) passes to the nonoxidative arm of PPP pathway to generate F6P and G3P, which re-enter glycolysis [31,32].

It is necessary to consider that chemically induced colitis models are often criticized for not accurately recapitulating ulcerative colitis in humans, a chronic inflammatory condition complicated by defects in innate and adaptive immunity and are considered merely models of acute inflammation [5]. Furthermore, it was acknowledged that most immune cells use specific and intrinsic metabolic reprogramming mechanisms to bioenergetically adapt while polarizing into a proinflammatory effector state [33]. With that in mind, our results may indicate the role of the measured activities of the respective enzymes in the pathways of primary carbohydrate metabolism: glycolysis, the PPP pathway, and the TCA cycle. Indeed, our results allow us to understand their adaptations in response to inflammatory stimuli in intestinal cells in colitis induced with TNBS. Furthermore, net serum lactate metabolites were quantified using an analysis of LDH, and LDH-A was determined using immunohistochemistry in colonocytes and leucocytes. For our results regarding the stain of LDH-A with immunohistochemistry, which was induced with a pro-inflammatory environment, the significant increase in LDH enzyme activity in leukocytes of the TNBS group agrees with previous studies that show that LDH-A activity promotes the expression of HIF-1α and My-c [34]. It is possible that LDH-A has a high glycolytic flux that exceeds maximal function during cell activation/proliferation, leading to the accumulation of pyruvate at toxic concentrations. A possible solution could be the expression of LDH-A, an enzyme that is able to efficiently remove pyruvate in the form of an organic acid, lactic acid, together with the regeneration of oxidised nicotinamide adenine dinucleotide (NAD^+^) [35]. The LDH activity of leukocytes from the lamina propria increased over the time, and there was also an increased lactate concentration, which was statistically significant in leukocytes. Moreover, lactate release by LDH-A promotes transcriptional modifications of IFN-γ through an epigenetic mechanism [36]. Although, evidence has shown that the rectal administration of lactate can downregulate the proinflammatory response of macrophages and dendritic cells since it is a potential beneficial microbiota metabolite and may contribute to health-promoting properties on the intestinal mucosa [37]. Recently, Zhang et al. [38] discovered that histone lactylation directly stimulates gene expression to promote M2-like characteristics (Arginase 1) in the late phase of M1 macrophage polarization. The deletion of LDH-A in M1 macrophages decreased M2 macrophage polarization. Our results for the activity of dehydrogenases as well as the Warburg effect as a substrate for the generation of lactyl-CoA for lysine lactylation on histones and their implications for IBD could be studied in more detail.

Regarding anion superoxide generation, our results provide in vivo evidence of increased significance in epithelium and lamina propria leucocytes in the TNBS groups. Similarly, the sites of inflammation tend to become depleted of molecular O_2_, nutrients, and metabolic intermediates and generate quantities of reactive nitrogen and oxygen species. These findings have led to the concept of ‘‘inflammatory hypoxia” [39,40]. One study proposed that under conditions of chronic hypoxia, electron transfer through flavoprotein channels electrons to ubiquinone (complex Q) to the respiratory chain [41]. This evidence could be related to the increased SDH enzymatic activity in both colonocytes and leukocytes of the TNBS group in our study. According to our results, leukocytes from rats treated with TNBS showed increased enzymatic activity of the pentose pathway and glycolysis, unlike epithelial cells. We also found that leukocytes with LDH-A switched to aerobic glycolysis. Translating this result to IBD, this increase in lactate may support the repair or mitigate the damage caused by the microbiota and the host’s immune system.

## Figures and Tables

**Figure 1 metabolites-13-00843-f001:**
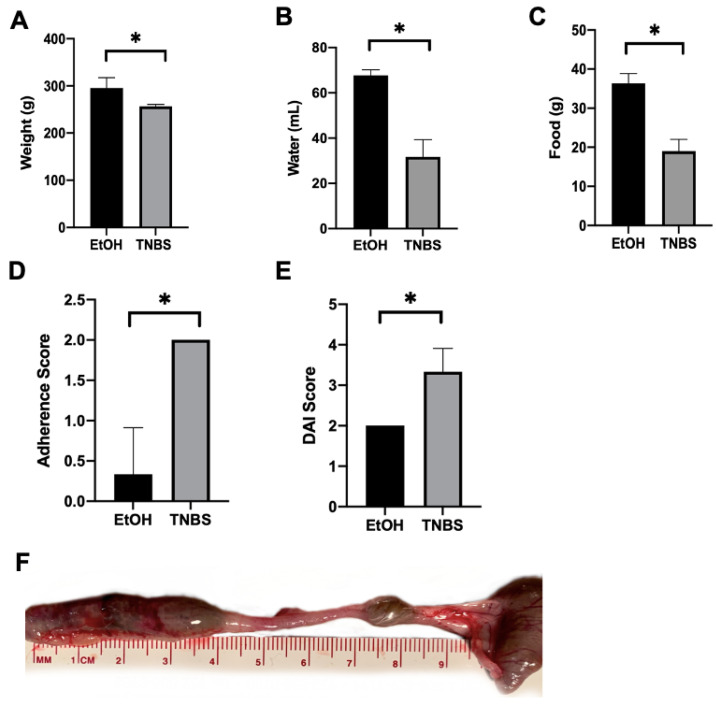
Parameters for the disease activity index score (DAI) related to (**A**) weight, (**B**) water ingestion, (**C**) food ingestion, (**D**) score, (**E**) score, (**D**) adherence, and (**F**) necrotic areas in EtOH and TNBS groups (* *p* < 0.05).

**Figure 2 metabolites-13-00843-f002:**
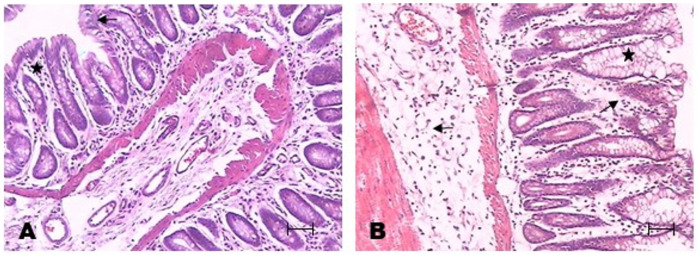
Colon epithelium haematoxylin and eosin (**A**,**B**). Colon samples from the EtOH group showed changes at the bottom of intestinal crypts (**A**), and oedema in the lamina propria in the TNBS group was observed ((**B**) arrow). Colon samples from the EtOH group showed changes at the bottom of intestinal crypts, * infiltrate (**A**).

**Figure 3 metabolites-13-00843-f003:**
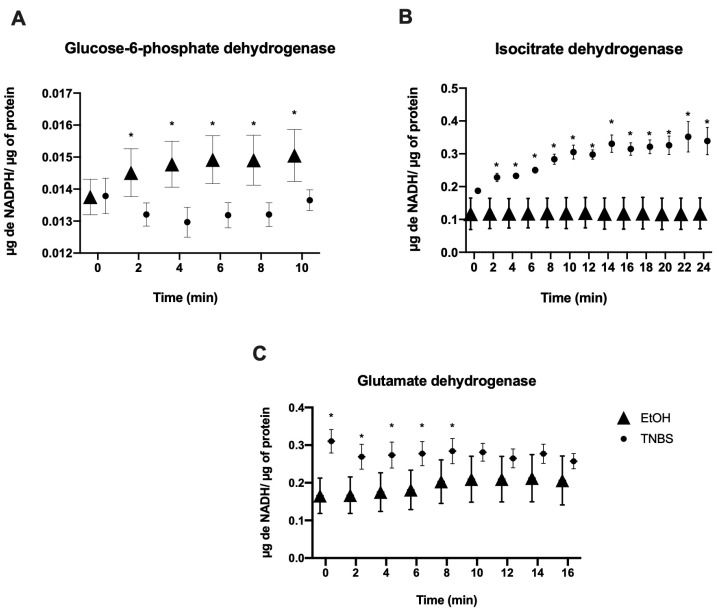
Enzymatic activity evaluation over time in the epithelial cells of rats. Epithelial cells from the EtOH and TNBS groups showed significant differences in different enzymatic activities. (**A**) Glucose-6-phosphate dehydrogenase (G6PDH), (**B**) isocitrate dehydrogenase (IDH), and (**C**) glutamate dehydrogenase (GDH) enzymes. Data represent the mean ± SD of three independent experiments (n = 6). *p*-values were determined using a one-way ANOVA (* *p* < 0.0001).

**Figure 4 metabolites-13-00843-f004:**
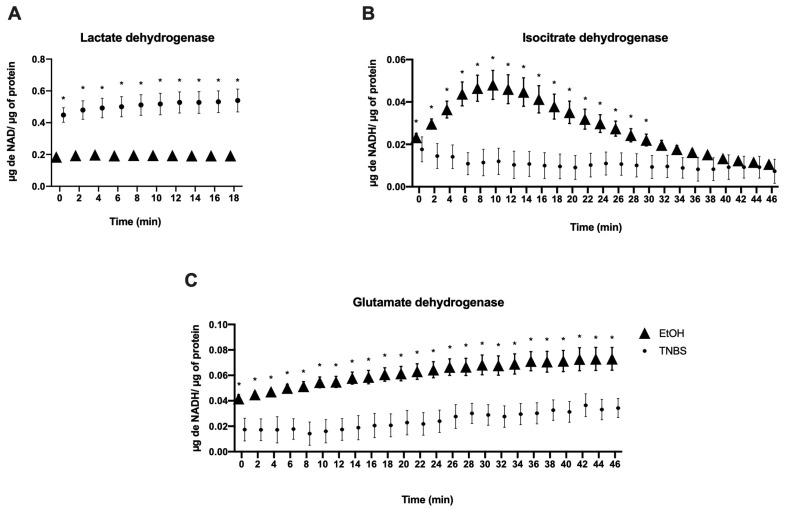
Enzymatic activity evaluation over time in leukocytes from the lamina propria. Leukocytes from the EtOH and TNBS groups showed significant differences for different enzymatic activities. (**A**) Lactate dehydrogenase (LDH), (**B**) isocitrate dehydrogenase (IDH), and (**C**) glutamate dehydrogenase (GDH) enzymes. Data represent the mean ± SD of three independent experiments (n = 6). *p*-values were determined using a one-way ANOVA (* *p* < 0.0001).

**Figure 5 metabolites-13-00843-f005:**
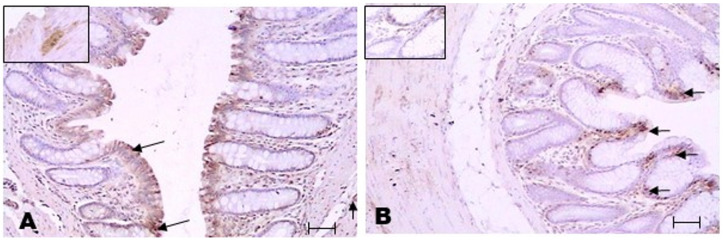
Colon epithelium LDH-A immunohistochemistry. Immunohistochemistry detected the presence of LDH-A in the colon epithelium of the EtOH group (**A**). LDH-A-positive leukocytes were observed in the inflammatory infiltrate in the TNBS group (**B**). Barr = 50 μm.

**Figure 6 metabolites-13-00843-f006:**
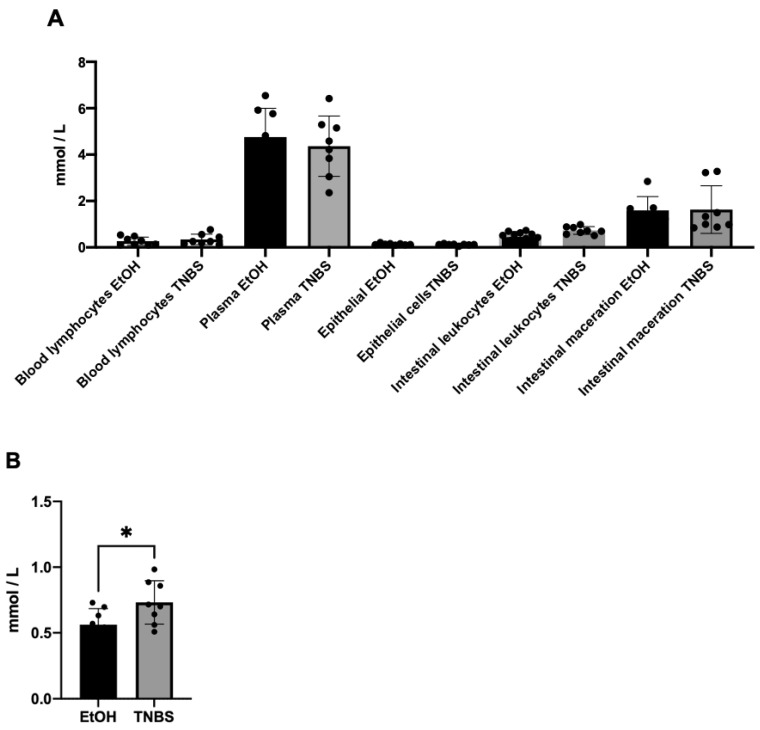
Lactate concentration. Quantification of released lactate. (**A**) The graphs show lactate release in mmol/L in blood lymphocytes, plasma, epithelial cells, intestinal leukocytes of lamina propria, and intestinal maceration from the EtOH and TNBS groups. (**B**) Lactate from leukocytes of EtOH- or TNBS-treated groups. A statistically significant difference is indicated with * (*p* = 0.001).

**Figure 7 metabolites-13-00843-f007:**
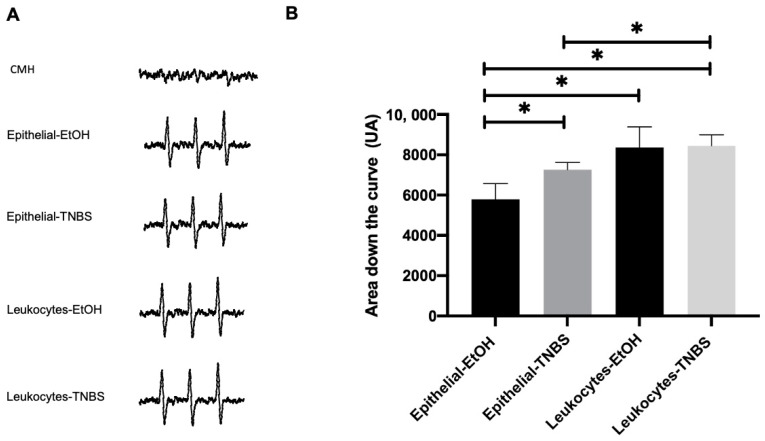
Quantification of superoxide anion using EPR. Different amounts of superoxide anion (•O_2_−) in epithelial cells and leucocytes of the EtOH and TNBS groups. (**A**) EPR spectrum indicating the presence of •O_2_− in the different study groups. (**B**) Graph showing the Y-axis as the area under the curve (arbitrary units) and the X-axis as different study groups (* *p* < 0.0001.).

**Table 1 metabolites-13-00843-t001:** Initial enzymatic activity from lamina propria leukocytes. G6PDH, GAPDH, LDH, and SDH showed significantly increased activities in leukocytes from the TNBS group vs. the EtOH group. Data represent the mean ± SD of three independent experiments (n = 6). *p*-values were determined using a one-way ANOVA.

Leukocytes from Lamina Propria			
	Initial Activity		Statistical Deference
	EtOH	TNBS	
G6PDH	0.019 ± (0.001)	0.035 ± (0.007)	*p* ≤ 0.01
GAPDH	0.232 ± (0.016)	0.657 ± (0.246)	*p* ≤ 0.04
LDH	0.185 ± (0.005)	0.448 ± (0.119)	*p* ≤ 0.01
IDH	0.023 ± (0.006)	0.018 ± (0.009)	*p* ≤ 0.447
GDH	0.042 ± (0.010)	0.017 ± (0.015)	*p* ≤ 0.082
SDH	0.011± (0.0008)	0.044 ± (0.0111)	*p* ≤ 0.006

## Data Availability

The data are contained within this article.

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
