# Peer review of "LDH-A Promotes Metabolic Rewiring in Leucocytes from the Intestine of Rats Treated with TNBS"

_metabolites, 2023, doi:10.3390/metabo13070843_

Round 1
Reviewer 1 Report
Please:
more detail introduction about IBDs, not just one sentence
list more recent references
more details in conclusions wih a comparasion with earlier research
Author Response
Reviewer 1
Please:
more detail introduction about IBDs, not just one sentence
R= We are very grateful to you for the careful reading of our manuscript. As you suggested we added more detail information in the introduction section, lines 47-62 and 69-97.
list more recent references
R= Thank you for your comment. There is scarce information on the subject, however, we have reviewed the bibliography and most recent references are in the manuscript.
more details in conclusions wih a comparasion with earlier research
R= Thank you for your observation. We have modified the conclusions and a comparison with earlier studies have made.

Reviewer 2 Report
The work by Mendoza-Arroyo and colleagues uses chemically induced colonic inflammation in a rat model. Addition of TNBS (2,4,6-trinitrobenzenesulfonic acid) results in the up- and down regulation of a series of metabolic enzymes, the accumulation of activated leukocytes and tissue damage. While the data are consistent, the manuscript is difficult to follow and in need of editing.
Results:
The subtitles need to be more informative For example “3.1 This Changes in the DAI score in TNBS-induced colitis in rats” better “A rise in the disease activity index (DAI) in the TNBS-induced colitis model”
All abbreviations & enzyme names need to explained again in the Result section the first time round.
The disease model needs to be explained again at the beginning of section 3.1 for the benefit of the reader.
Please explain the importance of “adherence”.
Please include a statement whether the TNBS induced ulcers, haemorrhage and necrotic areas are also seen in human patients.
Table 1 could be moved to Supplementary Materials
In Figure 2A is no asterisk although mentioned in the text.
Section 3.3: Start by explaining why the enzymes listed in Table 2 were selected. What is the rationale behind it? Please include a statement regarding the specificity of the used enzyme assays.
What is SDH?
…during the first 10 minutes after what? Please make it clear how the time lines are defined (I guess after addition of TNBS)
There is no conclusion at the end of the first paragraph in section 3.3. Generally speaking, all result sections should end with a conclusion highlighting the main finding leading to the subsequent section.
Fig3 & 4, please include the full name of the enzyme above each corresponding graph. Say in the figure legend how the time lines are defined.
Section 3.4: the text in the first section is redundant with Table 3. Better to edit the text so that it explains the data shown in Table 3 rather than repeating it.
Section 3.5: Please explain why LDH-A was selected for further analysis. Is LDH in section 3.4 identical to LDH-A? How specific is your assay for LDH-A?
Section 3.6: please explain to the reader why serum lactate was studied.
Legend Figure 6: it should say lamina propria leukocytes.
Make it clearer in the text that G6PDH changes differ in leukocytes compared to tissue.
Section 3.7: please explain why superoxide anions are investigated at this stage of the project.
Discussion:
make it clearer at the beginning of this section what the key observation are. Make it clearer that the changes in G6PDH are tissue/cell specific and explain why this matters more clearly. Refer to the relevant display items in the discussion. Explain more clearly the important roles of the underlying biochemical pathways (e.g. why is beta-oxidation important in this context?) in the discussion of the result to close the loop with the introduction.
Have you considered that the lactate-pyruvate balance may play an important role in the inflammation process? Especially lactate has received much attention recently as an epigenic & inflammation regulator.
Title:
I find the title too generic. Put your key message in the title and try staying under 100 characters (shorter titles have a higher click rate :))
Dear authors,
while the language needs moderate editing, the manuscript is in its current form difficult to follow. While most elemnts are there, the connecting narrative needs to be made more visible. Do this by including metacommentary (https://dhattengl110061fall2014.files.wordpress.com/2014/10/graff-metacommentary-reading.pdf) emphasising important observations while guiding the reader through the rational behind the experiments. Connect the result sections by concluding remarks. Put the take home messagy into title and subtitles. Always assume that the reader is no expert. Scientific writing is an argumentative communication between what the field knows & doen´t know, the results, the revierers/editor and the readership :)
Author Response
Reviewer 2
Comments and Suggestions for Authors
The work by Mendoza-Arroyo and colleagues uses chemically induced colonic inflammation in a rat model. Addition of TNBS (2,4,6-trinitrobenzenesulfonic acid) results in the up- and down regulation of a series of metabolic enzymes, the accumulation of activated leukocytes and tissue damage. While the data are consistent, the manuscript is difficult to follow and in need of editing.
R= We are very grateful to you for the careful reading of our manuscript. We have made modifications throughout the manuscript we hope now the manuscript is easier to understand.
Results:
The subtitles need to be more informative For example “3.1 This Changes in the DAI score in TNBS-induced colitis in rats” better “A rise in the disease activity index (DAI) in the TNBS-induced colitis model”
R= We thank the reviewer for his observation. We change the subtitles through the manuscript.
All abbreviations & enzyme names need to explained again in the Result section the first time round.
R=We thank the reviewer for your comment. As you requested, the enzymes abbreviations were again explained in the result section.
The disease model needs to be explained again at the beginning of section 3.1 for the benefit of the reader.
R= Thank you for your comment. Additional information of the disease model was added to section 3.1.
Please explain the importance of “adherence”.
R= Thank you for your observation. An explain of the adherence was added, line 325-326.
Please include a statement whether the TNBS induced ulcers, haemorrhage and necrotic areas are also seen in human patients.
R= Thank you for your comment. TNBS is only use in animals’ models to induce colitis, several papers have documented this.
- Morris GP, Beck PL, Herridge MS, Depew WT, Szewczuk MR, Wallace JL. Hapten-induced model of chronic inflammation and ulceration in the rat colon. 1989 Mar;96(3):795-803.
- Silva, I., Pinto, R., & Mateus, V. Preclinical Study in Vivo for New Pharmacological Approaches in Inflammatory Bowel Disease: A Systematic Review of Chronic Model of TNBS-Induced Colitis. Journal of clinical medicine, 2019; 8(10), 1574. org/10.3390/jcm8101574.
- Campbell, E. L., Bruyninckx, W. J., Kelly, C. J., Glover, L. E., McNamee, E. N., Bowers, B. E., Bayless, A. J., Scully, M., Saeedi, B. J., Golden-Mason, L., Ehrentraut, S. F., Curtis, V. F., Burgess, A., Garvey, J. F., Sorensen, A., Nemenoff, R., Jedlicka, P., Taylor, C. T., Kominsky, D. J., & Colgan, S. P. Transmigrating neutrophils shape the mucosal microenvironment through localized oxygen depletion to influence resolution of inflammation. Immunity, 2014, 40(1), 66–77. org/10.1016/j.immuni.2013.11.020
Table 1 could be moved to Supplementary Materials
R= Thank you for your observation. As you requested Table I has been moved to a supplementary figure.
In Figure 2A is no asterisk although mentioned in the text.
R= Thank you for your comment, we have added the asterisk to the figure 2A.
Section 3.3: Start by explaining why the enzymes listed in Table 2 were selected. What is the rationale behind it? Please include a statement regarding the specificity of the used enzyme assays.
R= Thank you for your comment. As you requested, we have added an explained related to the enzyme’s selection in introduction section and information regarding the specificity of the enzyme assays in material and methods.
What is SDH?
R= SDH is Succinate Dehydrogenase, now as you requested in a previous question, we have added in results the meaning of the enzyme abbreviations.
…during the first 10 minutes after what? Please make it clear how the time lines are defined (I guess after addition of TNBS)
R= Thank you for your observation. Now we have added information regarding the time lines, lines 384 and 385.
There is no conclusion at the end of the first paragraph in section 3.3. Generally speaking, all result sections should end with a conclusion highlighting the main finding leading to the subsequent section.
R= Thank you for your comment, we added the final conclusion at the end of the manuscript.
Fig3 & 4, please include the full name of the enzyme above each corresponding graph. Say in the figure legend how the time lines are defined.
R= As you requested, we have added the full names of the enzymes in each graph and in the figure legend, also the time lines was defined.
Section 3.4: the text in the first section is redundant with Table 3. Better to edit the text so that it explains the data shown in Table 3 rather than repeating it.
R= We thank the reviewer for this observation. We have edited the text in results 3.4.
Section 3.5: Please explain why LDH-A was selected for further analysis. Is LDH in section 3.4 identical to LDH-A? How specific is your assay for LDH-A?
R= We thank the reviewer for this comment. LDH and LDH-A are different enzymes, these are coded by different exon. LDH-A has a high glycolytic flux that exceeds maximal function during cell activation/biosynthetic process, leading to the accumulation of pyruvate to toxic concentrations is able to efficiently remove pyruvate in the form of an organic acid, lactic acid, together with the regeneration of oxidized nicotinamide adenine dinucleotide (NAD+), has been demonstrated in proliferating cells and biosynthetic process, favoring glycolysis an enzyme that is able to efficiently eliminate pyruvate in the form of an organic acid, lactic acid. This enzyme has also been found in immune cells. Lines 634-653. There not an specific assay for LDH-A.
1.Tannahill GM, Curtis AM, Adamik J, Palsson-McDermott EM, McGettrick AF, Goel G, Frezza C, Bernard NJ, Kelly B, Foley NH, Zheng L, Gardet A, Tong Z, Jany SS, Corr SC, Haneklaus M, Caffrey BE, Pierce K, Walmsley S, Beasley FC, Cummins E, Nizet V, Whyte M, Taylor CT, Lin H, Masters SL, Gottlieb E, Kelly VP, Clish C, Auron PE, Xavier RJ, O'Neill LA. Succinate is an inflammatory signal that induces IL-1β through HIF-1α. Nature. 2013 Apr 11;496(7444):238-42. doi: 10.1038/nature11986.
- Estévez-García IO, Cordoba-Gonzalez V, Lara-Padilla E, Fuentes-Toledo A, Falfán-Valencia R, Campos-Rodríguez R, Abarca-Rojano E. Glucose and glutamine metabolism control by APC and SCF during the G1-to-S phase transition of the cell cycle. J Physiol Biochem. 2014 Jun;70(2):569-81. doi: 10.1007/s13105-014-0328-1.
- Peng M, Yin N, Chhangawala S, Xu K, Leslie CS, Li MO. Aerobic glycolysis promotes T helper 1 cell differentiation through an epigenetic mechanism. Science. 2016 Oct 28;354(6311):481-484. doi: 10.1126/science.aaf6284.
Section 3.6: please explain to the reader why serum lactate was studied.
R= Thank you for your observation. The lactate was measured in plasma. In the manuscript a briefly explanation of plasma lactate was added.
Legend Figure 6: it should say lamina propria leukocytes.
R= Thank you for your observation, this was modified in the figure legend 6 A and B.
Make it clearer in the text that G6PDH changes differ in leukocytes compared to tissue.
R= Thank you for your comment. The G6PDH was only measured in epithelial cells and leukocytes isolated.
Section 3.7: please explain why superoxide anions are investigated at this stage of the project.
R= Thank you for your comment. We have added a briefly explanation about the investigation of superoxide anion, lines 104-105.
“In addition, we studied superoxide anion because it has been proposed that in some leukocytes it initiates signal transduction by increasing the production of superoxide anion (O2⨪) and hydrogen peroxide (H2O2) as second messenger”.
Gill T, Levine AD. Mitochondria-derived hydrogen peroxide selectively enhances T cell receptor-initiated signal transduction. J Biol Chem. 2013 Sep 6;288(36):26246-26255. doi: 10.1074/jbc.M113.476895
make it clearer at the beginning of this section what the key observation are. Make it clearer that the changes in G6PDH are tissue/cell specific and explain why this matters more clearly. Refer to the relevant display items in the discussion. Explain more clearly the important roles of the underlying biochemical pathways (e.g. why is beta-oxidation important in this context?) in the discussion of the result to close the loop with the introduction.
R= Thank for your observations. We have modified the discussion; we hope this clarify the relevant results of our work.
Have you considered that the lactate-pyruvate balance may play an important role in the inflammation process? Especially lactate has received much attention recently as an epigenic & inflammation regulator.
R= Thank for your observations. This comment is valuable because lactate-mediated histone lactylation and its implications for pathophysiology in some human diseases is under intense recent research. In this way, lactate is an epigenic & inflammation regulator.
- Zhang D, Tang Z, Huang H, Zhou G, Cui C, Weng Y, Liu W, Kim S, Lee S, Perez-Neut M, Ding J, Czyz D, Hu R, Ye Z, He M, Zheng YG, Shuman HA, Dai L, Ren B, Roeder RG, Becker L, Zhao Y. Metabolic regulation of gene expression by histone lactylation. Nature. 2019 Oct;574(7779):575-580. doi: 10.1038/s41586-019-1678-1
Title:
I find the title too generic. Put your key message in the title and try staying under 100 characters (shorter titles have a higher click rate :)
R= Thank you for your comment, we have modified the title of our work.

Reviewer 3 Report
The manuscript submitted to “Metabolites” an MDPI journal entitled: “Exploring metabolic pathways in epithelial and lamina propria leucocyte cells from the intestine of rats treated with TNBS” which discussed the enzymatic activity of glycolytic and Krebs cycle dehydrogenases and superoxide anions in 24 epithelial cells and leukocytes from the lamina propria in rats exposed to TNBS, the following points should be followed:
- Abstract should be rewritten in more details and high lighting the main results in order to sound better and giving strength to the manuscript.
- Introduction should be written with more details, especially with regard to TNBS.
- Introduction: at the end; add a paragraph about the aim of the work.
- Remove the word Exploring from the title of the manuscript.
- Materials and methods: more details should be added about the methodology not only depend on mentioning the references.
- Results were good, while discussion is very short and more details should be added as to discuss the whole obtained data.
- Add section for abbreviations.
- References: Should be updated till 2023.
Minor editing of English language required
Author Response
Reviewer 3
Comments and Suggestions for Authors
The manuscript submitted to “Metabolites” an MDPI journal entitled: “Exploring metabolic pathways in epithelial and lamina propria leucocyte cells from the intestine of rats treated with TNBS” which discussed the enzymatic activity of glycolytic and Krebs cycle dehydrogenases and superoxide anions in 24 epithelial cells and leukocytes from the lamina propria in rats exposed to TNBS, the following points should be followed:
- Abstract should be rewritten in more details and high lighting the main results in order to sound better and giving strength to the manuscript.
R= We are very grateful to you for the careful reading of our manuscript; we believe that your comments enrich our work. The abstract has been modified; we hope now our manuscript reflects the strength of the results obtained.
- Introduction should be written with more details, especially with regard to TNBS.
R= Thank you for your observation. The introduction was modified and details to TNBS was added, lines 50-56.
- Introduction: at the end; add a paragraph about the aim of the work.
R= Thank you for your comment, now we have added the aim of our work at the end of introduction, lines 98-105.
- Remove the word Exploring from the title of the manuscript.
R= Thank you for your observation. The title of the work was modified, and the word Exploring was deleted.
- Materials and methods: more details should be added about the methodology not only depend on mentioning the references.
R= Thank you for your comment. Material and methods have more detailed information to clarify.
- Results were good, while discussion is very short and more details should be added as to discuss the whole obtained data.
R= We thank the reviewer for this observation. Now the discussion has been modified, we added more details of whole data.
- Add section for abbreviations.
R= Thank you for your observation. According to authors instructions of this journal and the template, a list of abbreviations is not required but all abbreviations were described in the manuscript when were cited for first time.
- References: Should be updated till 2023.
R= Thank you for your comment, now new references have been added.
Comments on the Quality of English Language
Minor editing of English language required
R= The English language was reviewed by AJE, experts in the English editing, we added the certificate of English review.

Round 2
Reviewer 2 Report
Dear authors,
thank you very much for your thorough response to my queries. The readability of the manuscript is now much improved.
The English still needs some editing